# *Ferula* L. Plant Extracts and Dose-Dependent Activity of Natural Sesquiterpene Ferutinin: From Antioxidant Potential to Cytotoxic Effects

**DOI:** 10.3390/molecules25235768

**Published:** 2020-12-07

**Authors:** Roberta Macrì, Vincenzo Musolino, Micaela Gliozzi, Cristina Carresi, Jessica Maiuolo, Saverio Nucera, Miriam Scicchitano, Francesca Bosco, Federica Scarano, Stefano Ruga, Maria Caterina Zito, Lorenza Guarnieri, Ezio Bombardelli, Vincenzo Mollace

**Affiliations:** 1Institute of Research for Food Safety & Health IRC-FSH, University Magna Graecia, 88100 Catanzaro, Italy; micaela.gliozzi@gmail.com (M.G.); carresi@unicz.it (C.C.); jessicamaiuolo@virgilio.it (J.M.); saverio.nucera@hotmail.it (S.N.); miriam.scicchitano@hotmail.it (M.S.); boscofrancesca.bf@libero.it (F.B.); federicascar87@gmail.com (F.S.); rugast1@gmail.com (S.R.); mariacaterina.zito@studenti.unicz.it (M.C.Z.); lorenzacz808@gmail.com (L.G.); mollace@libero.it (V.M.); 2Nutramed S.c.a.r.l., Complesso Ninì Barbieri, Roccelletta di Borgia, 88021 Catanzaro, Italy; ezio.bombardelli@plantexresearch.it

**Keywords:** *Ferula* L., ferutinin, antioxidant and anti-inflammatory potential, phytoestrogenic activity, ionophoric property, mitochondrial dysfunction, ROS, NO, antiproliferative and cytotoxic activity

## Abstract

The employment studies of natural extracts in the prevention and treatment of several diseases highlighted the role of different species of genus *Ferula* L., belonging to the Apiaceae family, dicotyledonous plants present in many temperate zones of our planet. *Ferula communis* L. is the main source of sesquiterpene ferutinin, a bioactive compound studied both in vitro and in vivo, because of different effects, such as phytoestrogenic, antioxidant, anti-inflammatory, but also antiproliferative and cytotoxic activity, performed in a dose-dependent and cell-dependent way. The present review will focus on the molecular mechanisms involved in the different activities of Ferutinin, starting from its antioxidant potential at low doses until its ionophoric property and the subsequent mitochondrial dysfunction induced through administration of high doses, which represent the key point of its anticancer action. Furthermore, we will summarize the data acquired from some experimental studies on different cell types and on several diseases. The results obtained showed an important antioxidant and phytoestrogenic regulation with lack of typical side effects related to estrogenic therapy. The preferential cell death induction for tumor cell lines suggests that ferutinin may have anti-neoplastic properties, and may be used as an antiproliferative and cytotoxic agent in an estrogen dependent and independent manner. Nevertheless, more data are needed to clearly understand the effect of ferutinin in animals before using it as a phytoestrogen or anticancer drug.

## 1. Introduction

The use of herbal and plants extracts to prevent and to treat several acute and chronic diseases is an object of growing interest and investigation in pharmacological and nutraceutical research [1,2,3].

In vitro and in vivo evidence showed that the bioactive compounds isolated from herbs and plants showed antioxidant, anti-inflammatory and anti-cancer property [4,5,6]. 

The oxidative-stress reduction leads to a decreased risk of onset of several pathologies, such as cancer and chronic inflammatory diseases [7]. Among these the phenolic plants, classified into flavonoids, phenolic acids, stilbenes, tannins and lignans exert antioxidant activity in vivo and in vitro [8,9]; the stress oxidative reduction is due to their aromatic rings that contains hydroxyl groups and the antioxidant capacity depends on the number of these groups [10]. In particular, it has been observed that the flavonoid fraction of bergamot, an endemic plant of Calabrian region, showed pleiotropic effects reducing oxidative stress and cellular and tissue inflammation [11,12,13]. Furthermore, in vitro evidence suggests that another group of bioactive compounds, extracted from herbs, plants and fruits, such as monoterpenes, sesquiterpenes and diterpenes have antioxidant properties [14,15]. Indeed, the sesquiterpenes may play an important activity in human diseases, having anti-cancer [16], anti-inflammatory [17,18], bactericidal properties [19].

The results obtained in different experimental works suggest that the sesquiterpenes have protective potential at low doses and cause severe toxic effects at high doses [20]; this bipolar effect is due to the increases, induced by sesquiterpenes, of the cation permeability of lipid bilayers and mitochondrial membrane in a dose-dependent manner with a higher selectivity for divalent cations, such as calcium ions which play a key role in several pathophysiological processes [21,22,23]. 

Among the plants richest in sesquiterpenes and their derivatives, the genus *Ferula* L. is one of the most representative [24].

## 2. *Ferula communis* L.: Botanical Description and Chemical Composition

The genus *Ferula* L. belongs to the *Apiacae* family dycotiledonus plants comprising more than 400 genera and about 3700 species. This genus consists of about 170 species which mainly grow in the Mediterraneus zones, Northern Africa and Central Asia [24]. In Italy, three species have been described i.e., *Ferula communis* L., *Ferula glauca* L. and *Ferula arrigonii* Bocchieri. 

Particularly, in the Calabrian region, only one species, i.e., *Ferula communis* L., has been reported. *Ferula communis* L., vulgarly known as giant fennel is a perennial growing to about 3 m by 1 m herbaceous species [25]. The lower leaves are 3 to 4 pinnates, triangular, varied in size, soft, glabrous, green on both sides and usually have a conspicuous sheathing base. The lamina is finely divided into linear and filiform lobes. The latter have no distinct revolute margin and are up to 50 mm long, but no more than 1 mm. wide. The upper fertile leaves of the inflorescence are progressively reduced to a conspicuous sheathing base. The bracts are absent and the bracteoles are few or absent. The stem is very robust, wide (3–7 cm in diameter), full, finely striated and can grow to 2 to 3 meters high. The terminal fertile umbel is large and composed of 20 to 40 rays. The flower is bright yellow. The fruit (mericap) is elliptical or oblong-elliptical, strongly compressed dorsally; the length is varied between 7 and 15 mm [26,27]. 

This herbaceous plant presents different pharmaceutical activities related to different content of bioactive compounds extracted mainly from the roots, but also from the leaves and the rhizome. Indeed, there are two different chemotypes, which differs mainly for the produced effects. The toxic chemotype have a high percentage of ferulenol, a prenylated coumarin compound that leads to ferulosis, a lethal hemorrhagic syndrome in animals and humans [28]. The traditional use of non-toxic chemotype of *F. communis* L. is related to care of skin diseases, microbial and fungal infections, hysteria and dysentery [29].

The chromatographic analysis of non-toxic chemotype presents a high percentage of sesquiterpenes and their derivatives, such as ferutinin, lapiferin and teferin. The high concentration of ferutinin (ferutinol p-idroxybenzoate) is responsible for most of the effects produced by *Ferula* extract administration [30]. 

Ferutinin is a sesquiterpene, extracted from the roots, the leaves and the rhizome [31,32]. Different mechanisms by which several compounds, isolated from many species of *Ferula* L., exert their inhibitory activity on cell growth are known [33]. Among the compounds isolated from many species of *Ferula*, ferutinin obtained from the plants *Ferula ovina* Boiss., *Ferula communis* L., *Ferula hermonis* Boiss., and other species of *Ferula* [34,35], is known for the different activities that, in a dose-dependent way, performs both in vitro and in vivo (Figure 1). In particular, ferutinin has shown estrogenic, anti-inflammatory, antiproliferative, cytotoxic [36], antifungal and antimicrobial activities; the cytotoxic activity represents the pivotal role of its action that should leads to the use of this molecule in anti-neoplastic therapy [37,38,39,40].

## 3. Antioxidant Potential of Low Doses of Ferutinin

Low doses of ferutinin and other sesquiterpenes mimic the action of many antioxidant molecules, such as flavonoids and phenolic acids [8,9,41].

It has been demonstrated that low doses of ferutinin are associated with a strong antioxidant, anti-inflammatory and anti-tumorigenic activity [42,43], in fact the sesquiterpenes oxidize over time into sesquiterpenols, leading to a reduction of cellular oxidative status. In accordance, the results obtained in different experimental works suggest a key role of low doses of ferutinin in the reduction of free radicals production. This, in turn, determines an increase of antioxidant enzymes and glutathione and a decrease of inflammation [42,43]. 

In particular, an in vivo study demonstrated that products extracted from neuroactive endogenous medicinal plants, including ferutinin, are possible neuromodulators, with potential anticonvulsant properties at low doses. Indeed, these compounds, as observed for phenolic acids which in vitro and in vivo modulate the inhibitory action of glycine receptor, proved to be potential scavengers fighting against the range of neurodegenerative diseases by improving body antioxidant potential [43]. 

Raafat et al. (2015) studied the potential in vivo antioxidant and antihyperglicemic activity of *Ferula* extracts, including ferutinin at low doses (0.4–0.8–1.6 mg/Kg). At concentration of 1.6 mg/Kg, ferutinin was able to cause a significant glucose levels decrease and body weight increase in treated-mice respect to diabetic control group. Furthermore, consistent with previous experimental studies [44], it was observed that the same concentration of ferutinin administration significantly reverted the reduction trend of antioxidant enzyme catalase expression observed in diabetic mice. In addition to this evidence, this study for the first time described the antioxidant effects of ferutinin on neuropathic pain associated with diabetes, showing that 1.6 mg/Kg of ferutinin led to attenuation of tactile allodynia and thermal hyperalgesia [45].

## 4. Phytoestrogenic Activity of Ferutinin and the Hormone Replacement Therapy (HRT)

HRT is widely used to control menopausal symptoms and to prevent osteoporosis and dementia in women. However, its potential prothrombotic effects with a significant increased risk of venous thromboembolism and myocardial infarction led to evaluate the use of alternative estrogenic-therapy having the same efficacy, but with the reduction of side effects HRT-associated [46].

Several data highlighted that ferutinin could be an effective alternative hormone therapy to prevent and treat post-menopausal symptoms, since this sesquiterpene is able to mime the endocrine activity of the ovaries [47,48]. Thus, thanks to its similarity of structure with steroid hormones, ferutinin has been classified as a phytoestrogen with an affinity for both estrogen receptors subtypes ERα and ERβ and for the G protein-coupled estrogen receptor (GPER) [49]. 

ERα and ERβ receptors are transcribed from different genes and their expression is cell-type and tissue-type dependent. The nuclear estrogen receptors activate signalling pathway that result in the expression control of several genes by the direct binding to gene promoters or specific DNA sequences, called estrogen response elements (EREs). These palindromic sequences are located into the regulatory regions of target genes [50], such as the genes encoding the estrogen-responsive finger protein (EFP), which are strongly expressed in breast cancer, and the estrogen receptor-binding fragment-associated antigen 9 (EBAG9). Several transcription factors (TP53, FOS, JUN), secreted proteins (C3, AGT, LTF), proteins of intracellular signaling (BRCA1, BCL2, HRAS) are directly regulated by estrogenic action [51]. Instead, the GPER receptors are associated with a rapid cellular signaling, inducing cAMP production, ions mobilization, kinases activation, such as ERK1/2 and PI3K/Akt. In addition, GPER receptors regulates both calcium and potassium channels and are able to activate endothelial nitric oxide synthase (eNOS) with the subsequent maintenance of physiological nitric oxide (NO) production and, consequently, of endothelial function [52]. Overall, the possible activation of these mechanisms might justify the use of ferutinin as a more safe approach for the control of menopausal symptoms than “classical” HRT [47,48].

In vitro, the ferutinin binding affinity shows an inversion compared to other phytoestrogens since it has a greater affinity for estrogenic receptor ERα than for ERβ [49]; the in vivo phytoestrogenic activity confirms this binding specificity as, at the hypothalamic level, it has been observed a modulated expression of the ERα receptor [53]. It has been observed that ferutinin, when administered alone, increases the expression of ERα, while, when co-administered concomitantly with estradiol, decreases its expression [50,54]. Additional evidence demonstrated that ferutinin exerts an action comparable to estradiol benzoate, stimulating the endometrial proliferation. Moreover, ferutinin, dissimilar to estrogens, is able to increase the apoptotic index in luminal and glandular epithelia, avoiding the consequent stimulation of excessive cell proliferation leading to endometrial cancer. This represents an important goal because it might overcome the main risks related to the HRT [55].

In vivo studies also shown that ferutinin prevents osteoporosis and exhibits dose-dependent osteoprotective effects. Indeed, different doses of ferutinin exerted an important activity on bone remodelling in rats ovariectomized which reflects the effects of estrogen deficiency in menopause women [48,56,57]. In these experimental works the histomorphometric analysis of cortical bone of femoral diaphysis and of vertebral bodies, specifically the trabecular bone of lumen vertebrae and distal femoral epiphysis has been performed. The data obtained showed that the ovariectomized rats orally treated with ferutinin (2 mg/Kg/day for 60 days) showed the same bone mass values of ovariectomized rats treated with estradiol benzoate [54,56]. Another in vivo study conducted in ovariectomized rats, treated with the same concentration of ferutinin (2 mg/Kg/day for 60 days), it has been showed that ferutinin exerts an important action on body weight and bone loss, since the administration of this molecule is able to prevent together the excessive gain of weight and the osteoporosis caused by estrogen deficiency [56].

Furthermore, a recent study showed that ferutinin, through the regulation of Wnt/β catenin pathway, was able to promote osteogenic differentiation of dental pulp-derived stem cells, with a consequent promotion of bone density, acting on the promoter regions of Wnt Family Member 3A (*WNT3A)* and disheveled segment polarity protein 3 (*DVL3)*, by the activation of H3K9 acetylation and H3K4 trimethylation (not shown), upregulating their expression, as well as that of low-density lipoprotein receptor-related protein 6 (*LRP6*); instead ferutinin leads the downregulation of the expression of glycogen synthase kinase-3 (*GSK3)*. It is known that in the absence of *WNT, GSK3* constitutively phosphorylates the β-catenin and leads its ubiquitination and proteosomal degradation. However, in the presence of the ligand *WNT* and the binding of *WNT* protein to their receptor-co-receptor complexes (*Frizzled-LRP*), disheveled is activated and inhibits the phosphorylation of beta catenin by *GSK3*. The downregulation of *GSK3* expression is also associated with an inhibition of adipogenic process through the regulatory axis of *GSK3/STAT5/Sfrp/Wnt*. This evidence leads to hypothesize a potential role for Ferutinin as epigenetic activator and modulator of stem cells therapy to counteract osteoporosis (Figure 2) [58]. 

### Phytoestrogenic and Ionophoric Activity of Ferutinin: Effects on Platelet Aggregation

The ionoforic effects of ferutinin on platelets are in contrast to the other calcium ionophores; indeed, the standard ionophores produce a collapse of membrane ionic gradient in different cellular compartments, due to the calcium ions release from mitochondria, whereas ferutinin acts similar to an electrogenic uniporter and induces increased concentrations of cytosolic calcium, through the calcium release from endoplasmic reticulum, and a subsequent accumulation in mitochondria [59]. The intracellular calcium ions, in turn, increase plays a key role in different steps of platelet activation, firstly in the shape change and subsequently in platelet aggregation mechanisms [59,60,61]. 

In this context, the peculiar activity of ferutinin might be due to the association with the prominent affinity of ferutinin for estrogen receptors GPER, which are present on the platelet membranes [59,62,63]. In fact, the estrogenic regulation of platelets is independent of genomic regulation exerted by nuclear receptors ERα and ERβ, since the platelet are anuclear blood elements.

In this regard, the experimental studies conducted by Zamaraeva et al. (2010) demonstrated that the ionophoric activity that ferutinin exerts on calcium ions, going to up-regulate the latter, activating the fibrinogen receptors and increasing their binding, does not produce effects on spontaneous platelet aggregation [59,64,65]. 

Two hypotheses have been advanced to explain these results. The first regards the evidence that the ionophoric activity of ferutinin in platelet is able to stimulate the calcium-dependent NOS and the subsequent increase of NO; the NO elevation, through the cAMP and cGMP signaling, in turn, could directly and indirectly induce the calcium ions accumulation into the intracellular stores, reducing the calcium ions in platelets [59,66]. The second highlights that the cAMP level increase could follow the interaction with ferutinin and the estrogen receptor coupled to G-protein found on the platelet plasma membrane [59].

This evidence suggests that the activation of downstream intracellular signaling pathways of GPER within the platelet membrane could represent a potential target to develop new therapeutic strategies to prevent thrombosis and arterial ischemic events [67].

## 5. High Concentrations of Ferutinin: Pro-Oxidant and Cytotoxic Activity

At high concentrations, ferutinin triggers the overproduction of reactive oxygen species (ROS) in mitochondria, such as superoxide anion, at the level of the respiratory chain subunits I and III, through the opening induction of calcium channels and the subsequent alteration of the trans-membrane potential [36]. The increase in ROS produces different cell type dependent effects, both by promoting the oncogenes activation and the cancer initiation and progression [68], and by activating the intrinsic apoptotic pathway of caspase-9, producing an additive effect on cell death induction [69,70]. 

A growing number of data shows that many bioactive natural compounds lead to increase the ROS production and the related oxidative stress, with the subsequent activation of apoptosis pathway in tumor cell lines [71,72]. 

This increase seems to have a specific selectivity for the cancer cells compared to healthy cells: indeed, it has been observed a ferutinin-induced biphasic increase in intracellular calcium in T-leukemia cell line Jurkat. The plasma membrane of human lymphocyte expresses non-voltage-gated L-type calcium channels which have a crucial role in the biphasic calcium flux: the initial observed peak through the calcium mobilization from intracellular stores was maintained thanks the entry of extracellular calcium through the L-type calcium channels [61,73]. 

A recent study, using isolated rat liver mitochondria, aimed to evaluate the mitochondrial damages produced by ferutinin and demonstrated that the latter, was able to decrease the organelle respiration at concentration of 5–27 µM and to determine the membrane potential dissipation at concentration of 10–60 µM in a dose-dependent manner. It has also been observed that the mitochondrial effects of ferutinin might be primarily induced by stimulation of Ca^2+^-permeability, but different mechanisms, such as a driving of univalent cations, might be involved [74]. 

Furthermore, ferutinin, thanks to phytoestrogenic property, acts by enhancing the activity of NOS [75,76], through the increase of intracellular calcium [65]. The NOS presents three different isoenzymes that catalyse the production of the NO, a molecule that plays two divergent roles depending on the concentration [77,78]: in fact, the NO, at low concentrations, is involved in the cell viability processes performing its physiological functions at the neuronal, placental and endothelial level, while at high concentrations and for prolonged time, triggers cellular toxicity mechanisms going to activate different pathways of cell death, cell growth arrest and excitotoxicity [79]. 

The NO production, indeed, presents cell-specificity and dose-dependent effects [80], promoting apoptosis through mitochondrial dysfunction leading to release of pro-apoptotic factors, such as cytochrome c from the organelles [81]. 

Moreover, the NO bounds and inhibits the cytochrome c oxidase that leads to ROS overproduction, the consequent formation of peroxynitrite, that oxidizing and damaging irreversibly all mitochondrial complexes, affects the correct mitochondrial respiration triggering apoptosis (Figure 3) [82,83]. 

Increasing evidence on the ability of ferutinin to inhibit cell growth and to induce apoptosis both in vitro and in vivo [79] has led to a growing of research on ferutinin as an anti-cancer and chemopreventive agent, both alone and in combination with other chemotherapeutic drugs [84,85,86,87,88]. In particular, ferutinin had shown pro-apoptotic properties on several cell lines mediated by mitochondrial permeabilization and the release of molecules implicated in intrinsic apoptotic induction and progression. Among them, apoptosis inducing factor (AIF), a mitochondrial oxide-reductase, exerted a relevant role as physiologically participates in the electronic transport and, once it has released from the mitochondrion, moves into the nucleus acting as DNAse [89]. 

In addition, the SMAC/DIABLO molecule, inhibitor of the inhibitors of apoptosis proteins (IAPs), might be involved in the mechanism of ferutinin toxicity, as once leaked from the mitochondria, it can bind IAP, preventing the inhibition of caspase activity and promoting the formation and activation of the apoptosome complex, leading to caspase-9 activation [90,91] that is considered an initiator of a cascade of caspases. The effector caspases 6 and 7, in turn, once activated, cleave different substrates, such as laminin, actin, vimentin and ICAD (Figure 4) [92], causing apoptotic cell death.

Among cell lines tested to verify ferutinin effects, it was observed that, for tumor cell lines, human breast adenocarcinoma cell line (MCF-7), human cells of urothelial carcinoma (TCC), human colon adenocarcinoma cells (HT29), murine cells of colon carcinoma (CT26), IC 50 was between 67 and 81 μM, while the IC 50 for normal human and murine fibroblast lines (HFF3 and NIH/3T3) was higher, with values of 98 and 136 μM respectively, with about 67% of viable cells in both cell lines treated at the above concentrations (Table 1) [32,33,34,35,36]. Another study conducted by Alkhatib et al. (2008) showed that the ferutinin produced cytotoxic effects on (imatinib-resistant) human chronic myeloid leukemia cell line (K562R) and (dasatinib-resistant) mouse leukemia cell lines (DA1-3b/M2BCR-ABL), respectively at IC 50 25.3 μM and 29.1 μM [93].

Furthermore, Arghiani et al. (2015) observed cytotoxic effects of ferutinin on human teratocarcinoma (NTERA2) and oesophageal cancer (KYSE30) cell lines and the IC50 values were respectively 39 μM and 58 μM [94].

The anti-cancer ferutinin activity was also evaluated in vivo, in comparison with cisplatin, on tumors developed in hybrid BALB/C mice: a tumor reduction of 67% for ferutinin and of 72% for cisplatin was observed; however while the latter produced a significant alteration of the liver tissues and the spleen of the treated mice, this did not happen for mice in treatment with ferutinin [95]. 

In vitro studies have shown that, thanks to its phytoestrogenic activity, ferutinin induced the inhibition of proliferation in human breast carcinoma MCF-7 cells [96]. To test whether ferutinin actually acts as a protective agent on breast and uterine carcinoma, two different mechanisms involved in the initiation and in the progression of cancer, respectively, were assessed. The first was the evaluation of the cell proliferation rate through the Ki-67 antigen, a non-histone nuclear protein used as a good proliferation marker, as it is present at low levels in the quiescent cells and increases in proliferating cells, especially in phase G2, M and in the second half of phase S. The second was the detection of DNA fragments generating during the apoptotic process [97]. These studies showed that ferutinin produced dose-dependent effects and, in particular, the oral dose of 2 mg/kg/day offered the best protective action, increasing the rate of apoptosis in the glandular epithelia of ovariectomized rats [97]. In addition, ferutinin showed its antiproliferative activity on human and murine leukemia cell lines [93].

A recent study confirmed the cytotoxic activity of ferutinin also on the prostate cancer (PC-3) cell-line, with IC 50 values of 16.7 µM. The low IC 50 is due to the cross-link among the cytotoxic activity and hormonal action, since the prostate cancer onset is primarily due to androgenic hormones [98].

Gao et al. (2013) showed that ferutinin, in a dose-dependent way and, particularly, at high concentrations (40 μM or above), was able to induce apoptosis in human red blood cells (RBCs), without the involvement of mitochondria since these cells lose those organells during the maturation process. The erythroptosis is mediated by simultaneous increase of the cytosolic free calcium ions level and the activity of caspase-3. In addition, the external calcium depletion had not blocked the apoptosis mediated by ferutinin, suggesting that the existence of unknown mediators of apoptosis are ferutinin-induced, which could be added to the ionophoric activity [99]. Certainly, this toxicity of ferutinin in vitro suggests that more studies on animals are needed to examine effect of ferutinin before evaluating it as a phytoestrogen or anticancer drug.

## 6. Conclusions and Futures Perspectives

The *Ferula* genus extract contains a high percentage of the bioactive compound ferutinin, having a wide range of biological effects that elect it as a possible therapeutic agent with a broad spectrum of action. We examined the effects of ferutinin, highlighting the deep gap existing between the uses of high and low doses.

Indeed, the ferutinin administration at low-concentration is associated with an important antioxidant action, although the mechanisms of free radical decrease still need further clarification.

Moreover, the effects of cross-talk between the estrogenic and ionophoric properties of ferutinin that lead to strongly reduction of the side effects produced by usual estrogen therapy, allow this molecule to be eligible as a potential therapeutic agent in the gynecological pathologies. 

Concerning the antiproliferative action of ferutinin, several in vitro and in vivo studies showed that the bioactive compound acts both on cytotoxicity and cell proliferation in an estrogen-dependent and -independent manner and its activity is dose dependent and variable according to the cell type analysed, with a preferential action for tumor cell lines.

The ionophoric effect of ferutinin with the consequent loss of the trans-membrane potential seems to be the key point of the antiproliferative and cytotoxic effect, both directly through the activation of mitochondrial-dependent apoptosis and indirectly through the effects produced by the summarization of NO and ROS, molecules that play a key role in the production of stress, senescence and death signals.

Therefore, based on the evidences obtained, the bioactive compound ferutinin can be considered an effective therapeutic agent as adjuvant or substitute for the current anti-neoplastic treatments. More studies should be performed to evaluate its action on other cell types and further molecular pathways involved, to compare its activity with that of chemotherapy drugs going to evaluate the different degree of toxicity produced as a side effect on healthy cells.

## Figures and Tables

**Figure 1 molecules-25-05768-f001:**
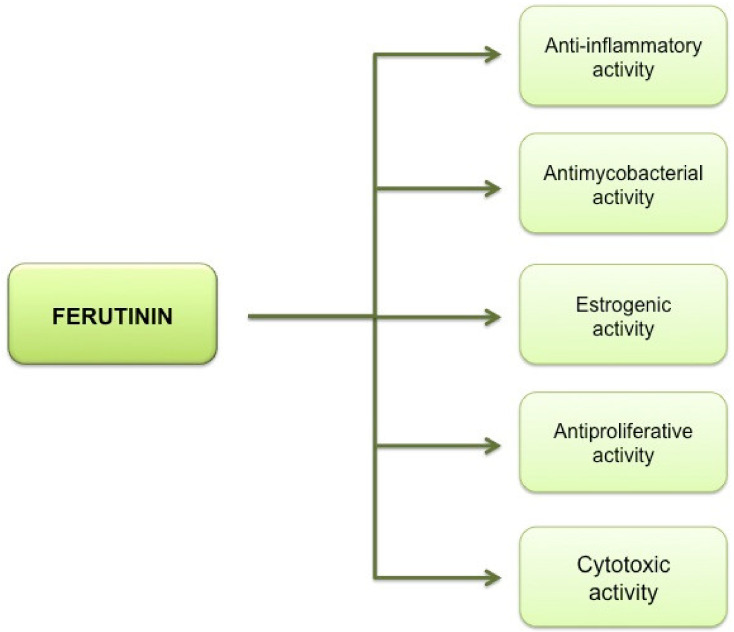
The pleiotropic activities of sesquiterpene ferutinin. Ferutinin exerts several activities in human diseases, having estrogenic, anti-cancer, anti-inflammatory and bactericidal properties.

**Figure 2 molecules-25-05768-f002:**
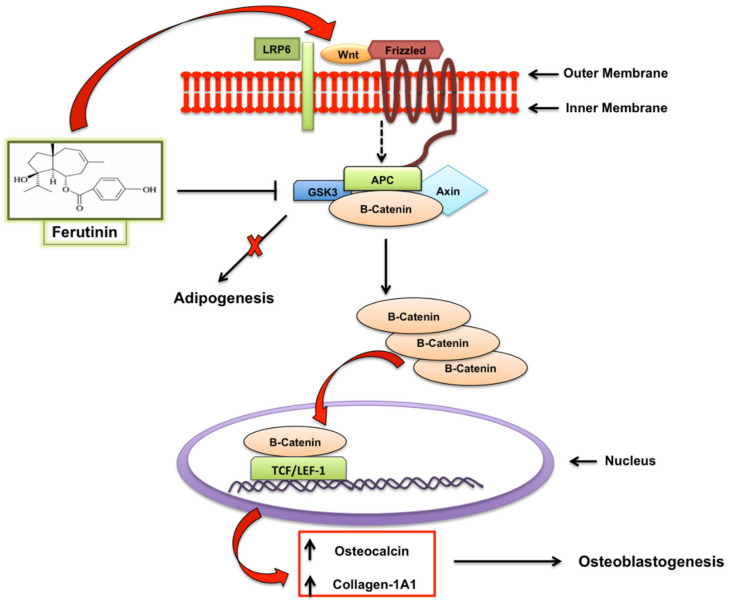
Regulation of Wnt/β catenin pathway by ferutinin: ferutinin induces osteogenic differentiation of dental pulp-derived stem cells and leads to the promotion of osteoblastogenesis.

**Figure 3 molecules-25-05768-f003:**
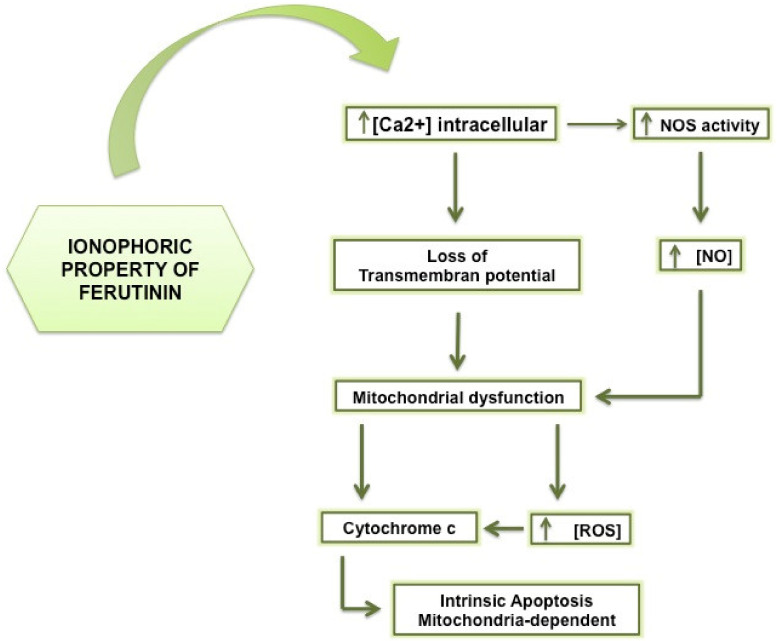
The ionophoric property of sesquiterpene ferutinin. Ferutinin, thanks to its ionophoric activity leads to loss of transmembrane potential and the subsequent apoptosis induction.

**Figure 4 molecules-25-05768-f004:**
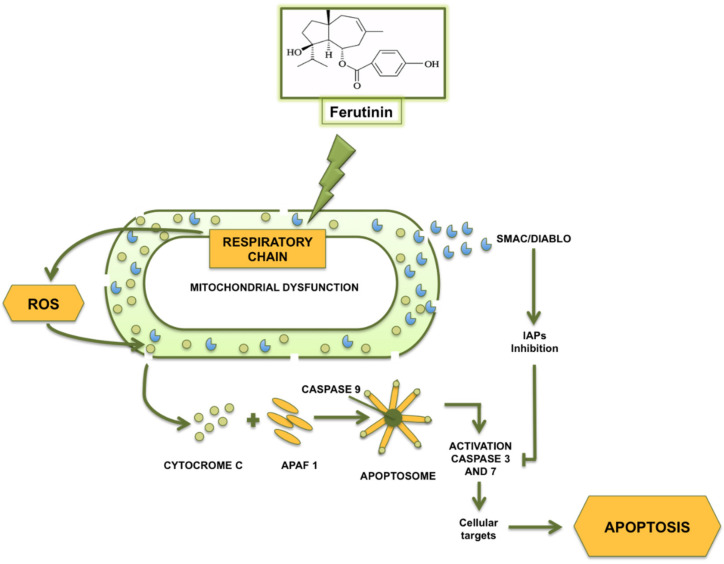
Ferutinin-induced mitochondrial dysfunction: key role of ferutinin action on respiratory chain complex I and III.

**Table 1 molecules-25-05768-t001:** IC50 values of different types of cell lines, tumor and healthy respectively, treated with ferutinin.

Cells Treated	IC 50	Ref
**MCF–7:** Human breast adenocarcinoma cell line	81 μM	[32]
**TCC:** Human Transitional Cell Carcinoma Cell line	67 μM	[32]
**CT26:** Murine colorectal carcinoma cell line	81 μM	[32]
**HT29:** Human colorectal adenocarcinoma cell line	72 μM	[32]
**K562R:** Human chronic myeloid leukemia cell line	25.3 μM	[93]
**DA1-3b/M2BCR-ABL:** Mouse leukemia cell line	29.1 μM	[93]
**NTERA2:** Human teratocarcinoma cell line	39 μM	[94]
**KYSE30:** Human oesophageal cancer cell line	58 μM	[94]
**PC3:** Prostate cancer cell line	16.7 μM	[98]
**HFF3:** Human foreskin fibroblast cell line	98 μM	[32]
**NIH/3T3:** Mouse embryo fibroblast cell line	136 μM	[32]

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
