# Peer review of "Ferula* L. Plant Extracts and Dose-Dependent Activity of Natural Sesquiterpene Ferutinin: From Antioxidant Potential to Cytotoxic Effects"

_molecules, 2020, doi:10.3390/molecules25235768_

Round 1
Reviewer 1 Report
Ferutinin administration at low-concentration is associated with an important antioxidant action and high concentration and cyto-toxic effect need to be emphasized in milder word. This two point contradict each other. I would say highlight issues more precisely and/or in the context of opportunity (cytotoxic effect) for therapeutic intervention.
Ferutinin induces osteogenic differentiation need more reference beyond WNT signaling. Osteoclast specific examples such as RANKL, Pit formation, microCT animal data, bone health context will be more meaningful here.
Talk about bioavailability of Ferutinin.
Reviewer 2 Report
This review article gives a detailed overview of the positive effects and possible use of ferutinin from the genus Ferula. In addition, the authors have described all activities of ferutin in detail, with particular emphasis on the graphical representations (Figure 2-4). The authors also make use of the relevant scientific literature (up-to-date), which describes in detail the molecular mechanisms involved in the various activities of ferutinin.
I recommend the publication of the paper as it is of significant importance for future researches especially in the field of bioactive compounds significance and applications.
Author Response
We are grateful to the reviewer for appreciating our work.
Reviewer 3 Report
This is a review of the multiple effects of Ferutinin.
- The font too small in Figures 2 and 4
- In Figure 2 - what part does Ferutinin inhibit? Does it easily cross the membrane to get intracellular? In the diagram it looks like it activates and inhibits the same group of proteins. Which is it? What experiments helped identify this? The paragraph describing this figure - lines 161-166 don’t clearly describe where ferutinin plays a role in this pathway as drawn in the figure. H3K9 and H3K4 are mentioned in the text but aren’t in the diagram.
- All abbreviations should be spelled out even if only appearing in the diagram and even if just listed as transcription factors involved.
- Numbers should be standardized and use a decimal instead of a comma to be consistent with scientific writing.
- Line 150-152 - is this in humans or animals or cells?
- Some spots have double spacing after a word (line 129, 188… )
- Line 235 - spelling of “transmembran” should be “transmembrane”
- Cite the article in the table 1 legend/description since you are getting these numbers from a different paper. Really if you are doing a review a whole data table shouldn’t come from 1 source. You should be combining information from a few sources to “review” what is out there not just make a table of 1 other papers data. I recommend adding more data and moderately exapanding table 1.
- Punctuation line 286 - is a period needed after Ferutinin? Instead of a comma?
- If Ferutinin causes apoptosis in rbc’s how does that limit its potential as a treatment?
- Line 99 - what do you consider to be low dose? Throughout the paper - You have some doses in ug/ml, mg/kg and uM. You cannot easily compare those… The same unit should be used throughout for cells and then another unit throughout for animal exps.
- Do you hypothesize treating locally or systemically… how were the animals treated in experiments mentioned? Intraperitoneal or orally?
- Line 309 - “More studies must be performed” is strong - “More studies should be performed” may be more accurate.
